# Causal Inductive Synthesis Corpus

**Zenna Tavares**
Massachusetts Institute of Technology
Cambridge, MA 02139
zenna@mit.edu

**Ria Das**
Massachusetts Institute of Technology
Cambridge, MA 02139
riadas@mit.edu

**Elizabeth Weeks**
Massachusetts Institute of Technology
Cambridge, MA 02139
eweeks@mit.edu

**Kate Lin**
Wellesley College
Wellesley, MA 02481
kate.lin@wellesley.edu

**Joshua B. Tenenbaum**
Massachusetts Institute of Technology
Cambridge, MA 02139
jbt@mit.edu

**Armando Solar-Lezama**
Massachusetts Institute of Technology
Cambridge, MA 02139
asolar@csail.mit.edu

## Abstract

We introduce the Causal Inductive Synthesis Corpus (CISC) – a manually constructed collection of interactive domains. CISC domains abstract core causal concepts present in real world mechanisms and environments. We formulate two synthesis challenges of causal model discovery: the passive discovery of a model of a CISC domain from observed data, and active discovery while interacting with the domain. CISC problems are expressed in AUTUMN, a Turing-complete programming language for specifying causal probabilistic models. AUTUMN allows succinct expression for models that vary dynamically through time, respond to external input, have internal state and memory, exhibit probabilistic non-determinism, and have complex causal dependencies between variables.

## 1   Introduction

Young children engage in forms of intuitive scientific discovery, building structured causal theories of their environment [13, 3, 2] using many of the principles that underpin professional science. Significant progress has been made in modelling many of these principles – in particular Bayesian inference over structured representations [15, 5, 12], discovery of causal models from observational and interventional data [14, 4, 9], and the optimal design of experiments [11, 6, 10, 7]. Nevertheless, automatic discovery of models of realistic phenomena from observation and interaction remains largely out of reach. The objective of this contribution is to facilitate progress towards automatic scientific discovery, first by introducing a representation of causal models that is expressive enough to succinctly capture the complexities of real world phenomena, and second by presenting a corpus of domains and accompanying benchmark challenge.

Most real-world causal mechanisms are complex. They often possess internal state, have time-varying behaviour, and are composed of both continuous and discrete components with complex logical and algorithmic relationships between them. For instance, a typical microwave will heat only if the door is closed, a duration has been keyed in, and the start command has been pressed. Radiation then causes a continuous increase in temperature, while the display discretely counts down the remaining time.

34th Conference on Neural Information Processing Systems (NeurIPS 2020), Vancouver, Canada.

```
-- define Ants and Food
object Ant {(Cell 0 0 gray)}
object Food {(Cell 0 0 red)}

-- ants initially randomly placed
ants : List Ant
ants = init map Ant (randPositions GRID 6)
       next update (prev ants) nextAnt

-- food gets removed if ant lands on it
foods : List Food
foods = init ()
        next update (prev foods)
                    obj -> if nextTo obj (closest obj Ant)
                           then removeObj obj
                           else obj

-- add random food to grid on click
on clicked
   foods = addObj (prev foods)
                  (map Food (randPositions GRID 4))

-- move every ant to the closest food
nextAnt : (Ant -> Ant)
nextAnt ant = move ant (unitVec ant (closest ant Food)))
```

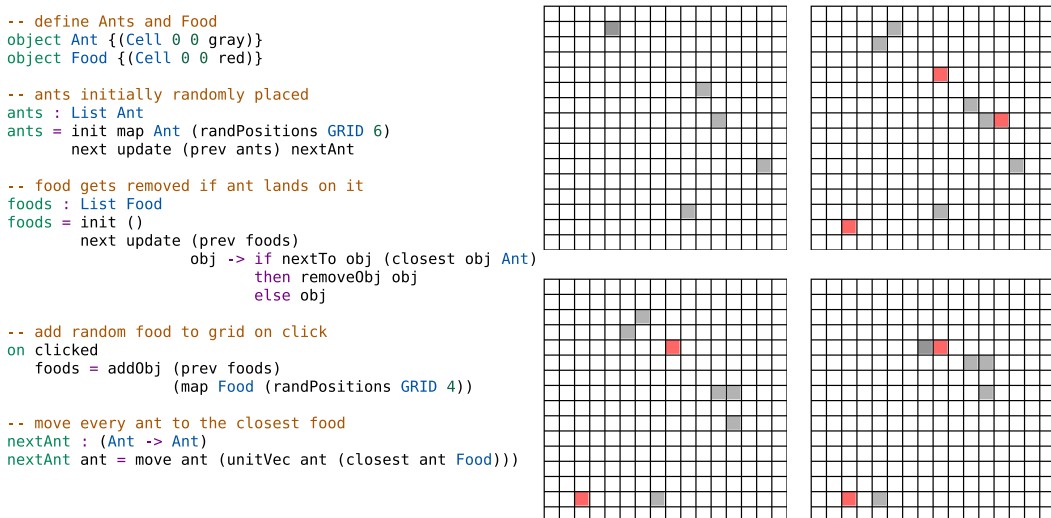

Figure 1: An AUTUMN program. This program simulates ants seeking food, starting at $t = 0$ from top-left, clockwise. A number of ants (grey) are initially randomly positioned on the grid. On clicking, food (red) is placed at random positions on the board. All ants then take a move at each timestep towards the closest food item.

These properties are difficult or impossible to represent them with traditional modelling formalisms, such as causal graphical models.

Complex mechanisms can, on the other hand, be easily encoded as programs – indeed many (such as the internal logic of a microwave) are literally programs. We introduce a functional reactive language [1, 8] called AUTUMN for expressing causal probabilistic models. Functional reactive programming languages augment functional languages – which are oriented around defining pure mathematical functions – with primitive support for temporal events. AUTUMN allows succinct expression of models that vary dynamically through time, respond to external input, have internal state and memory, exhibit probabilistic non-determinism, and have complex causal dependencies between variables.

Within a programmatic representation of causal models, such as AUTUMN, scientific discovery is then in part a problem of program synthesis. However, unlike conventional synthesis formulations, causal direction is important and must be inferred. Moreover a complete model of scientific discovery cannot presuppose that the necessary data is given; what experiments to run and what data to collect are integral parts of the problem. We introduce the Causal Inductive Synthesis Corpus (CISC), a suite of interactive models designed for causal discovery. CISC domains are distillations of mechanisms, modelling causal concepts such as circuits, magnetism, disease propagation, fluids, and foraging. CISC problems resemble simple video games but have no notion of external reward. Instead, CISC provides a framework for defining challenging passive and active inductive synthesis problems.

## 2 The Autumn Language

AUTUMN is designed to express models that vary as a function of time or external input. Many constructs are standard. The expression x = val binds the symbol x to the value val. Local values are bound using **let**. Values have types; x : Int denotes that x is of type Int. Functions are values with function types. f : Int → Bool denotes the type of functions the Int to Bool.

**Sequences** Values in an AUTUMN program represent sequences that vary with time. A value $v$ at time $t$ may be (i) time invariant, i.e., $v_t = c$ for some constant $c$, (ii) stateless and time varying, i.e, $v_t = f(t)$ for some function $f$, or (iii) stateful / recurrent sequences defined in terms of previous values, i.e., $v_t = f(v_{t-1})$. In AUTUMN, sequences are specified by defining two expressions in a recurrence relation: the initial value and the value as a function of previous values. These are constructed using a primitive language pattern **init** expr1 **next** expr2, which is sufficient to express the aforementioned kinds of sequences:

$$\begin{array}{rl}
\text{Variables} & x, y, z \in \text{Var} \\
\text{Type } \tau ::= & \text{Int} \mid \text{Bool} \mid \text{Real} \mid \tau_1 \to \tau_2 \mid \Omega \\
\text{Term } t ::= & n \mid b \mid r \mid \bot \mid x \mid \lambda x : \tau.t \mid \\
& \textbf{if } t_1 \textbf{ then } t_2 \textbf{ else } t_3 \mid t_1 \oplus t_2 \mid \\
& t_1 \; t_2 \mid \textbf{let } x = t_1 \textbf{ in } t_2 \mid \\
\text{(temporal terms)} & \textbf{on } t_1 \; t_2 \mid \textbf{init } t_1 \textbf{ next } t_2 \\
\text{prob terms} & \textbf{rand}
\end{array}$$

Figure 2: Abstract Syntax

1. Time invariant values are simply constants v = **init** 3 **next** 3. This can also be expressed more succinctly as simply v = 3.

2. Stateless, time-varying values are simply functions of time:

    ```
    v = init iseven time
          next iseven time
    ```

    This can also be expressed more succinctly as simply applying a function to an existing time varying value, which is interpreted pointwise:

    ```
    v = iseven time
    ```

3. Stateful and recurrent sequences refer to previous values in their definition. The simplest example is perhaps time itself, which need not be defined as a primitive, but can be expressed using the primitive **prev** x which returns the value of x at the previous timestep:

    ```
    time = init 1
            next (prev time) + 1
    ```

    A more complex example is a value that evolves according to the Fibonnaci sequence:

    ```
    fib = init 0
           next if time == 1
               then 1
               else (prev fib) + (prev prev fib)
    ```

**Events**  A second way to specify temporal events is using the construct **on**, with the pattern **on** event intervention. An event is any sequence of type Bool, and an intervention is a modification to a value. In the following example, if a click occurs at a time later than 5, the value of x will reset to 0:

```
x = init 0 next (prev x) + 1
on click & (time > 5)
   x = 0
```

The primitives **on**, **init**, **next**, **prev** in combination with a standard library enable succinct expression of a wide variety of models (see Figure 1).

## 3 The Causal Inductive Synthesis Corpus

The Causal Inductive Synthesis Corpus (CISC) is a collection of environments expressed within the AUTUMN language. CISC problems are abstract models of mechanisms and environments.

**Specification**  Let $\mathcal{L}$ denote the set of all AUTUMN models. CISC is a dataset $\mathcal{D} = (m_1, m_2, \ldots, m_N)$ of $N$ AUTUMN models, i.e., $m_i \in \mathcal{L}$. For each model $m \in \mathcal{D}$, there is also a collection of $M_m^{\text{test}}$ test trajectories $T_m^{\text{test}} = (\tau_1, \tau_2, \ldots, \tau_{M_m^{\text{test}}})$ and $M_e^{\text{train}}$ train trajectories $T_m^{\text{test}} = (\tau_1, \tau_2, \ldots, \tau_{M_m^{\text{train}}})$. A trajectory is a pair $(\boldsymbol{a}, \boldsymbol{o})$, where $\boldsymbol{a}$ and $\boldsymbol{o}$ are finite sequences (of identical length) of actions and observations respectively. The action space $\mathcal{A} = \mathbb{N}^2 \cup \{\uparrow, \downarrow, \leftarrow, \rightarrow, \text{skip}, \text{stop}\}$ allows for selecting a grid-cell, pressing an arrow, performing no action, or stopping a simulation. The observation space $\mathcal{O} = C^{W \times H}$ is a colored grid of cells, where $W$ (width), $H$ (height) are constants and $C$ is a set of colors.

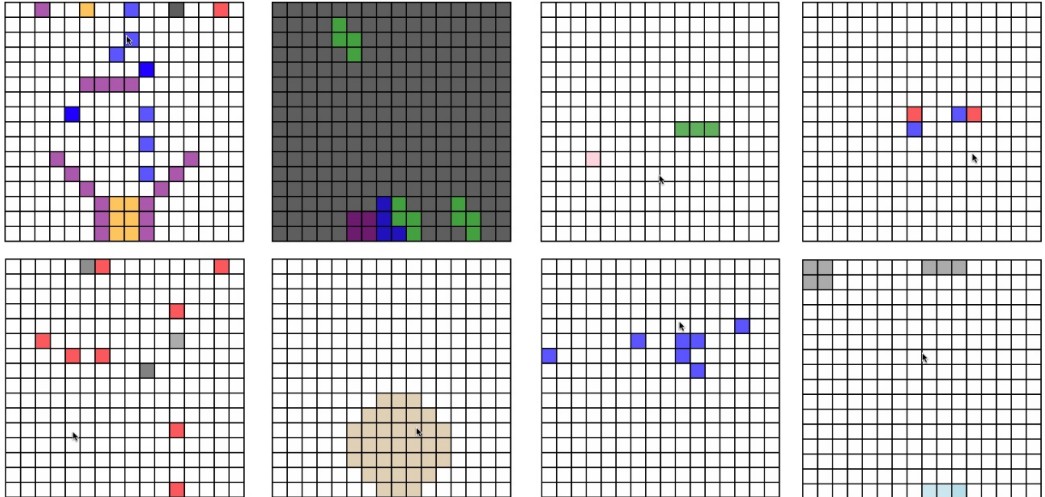

Figure 3: Example domains from Causal Inductive Synthesis Corpus. From top-left clockwise: a simulation of water interacting with a sink, a Tetris clone, a snake clone, interacting magnets, food-seeking ants, obfuscated objects, a particle simulation, a simple weather simulation.

**Passive Discovery:** The passive inductive synthesis problem is to produce a synthesizer $s$ that maps a set of trajectories $T_m^{\text{train}}$ produced from a ground truth AUTUMN model $m$ onto a hypothesis AUTUMN model $\hat{m} = s(T_m^{\text{train}})$ where $m, \hat{m} \in \mathcal{L}$.

The score of a hypothesis $\hat{m}$ is a measure of the degree to which it matches $m$ on the test trajectories. Recalling that AUTUMN programs may be probabilistic, let sim denote a stochastic simulation function such that $\text{sim}(m, \boldsymbol{a})$ is a random variable over observations, the score of $m$ is marginal likelihood averaged over $T_m^{\text{test}}$:

$$\text{score}_m(\hat{m}) = \frac{1}{M_m^{\text{test}}} \sum_{(\boldsymbol{a}_i^m, \boldsymbol{o}_i^m) \in T_m^{\text{test}}} p(\text{sim}(\hat{m}, \boldsymbol{a}_i^m) = \boldsymbol{o}_i^m) \tag{1}$$

The score of a synthesizer $s$ is then the average score over $\mathcal{D}$:

$$\text{score}(s) = \frac{1}{N} \sum_{m \in \mathcal{D}} \text{score}_{m_i}(s(T_m^{\text{train}})) \tag{2}$$

m

**Active Discovery** In contrast to the passive case, in active discovery the observational data is not given and must be produced by an active agent. The active inductive synthesis problem is to produce a pair $(\pi, \sigma)$ where $\pi : \mathcal{O} \times \Phi \rightarrow \mathcal{A} \times \Phi$ is a policy with internal memory $\Phi$, and $\sigma$ is stateful synthesizer. The agent interacts with a model, producing observational data until a `stop` action is performed. At this point a hypothesis model $\hat{m} = \sigma(T, \phi)$ is produced as function of the internal state $\phi \in \Phi$ of the agent and the trajectory $T = (\boldsymbol{a}, \boldsymbol{o})$ it has observed.

For the sake of evaluation consistency we force the domains to be deterministic by fixing the random seed, and hence $T$ is a function of a model $m$ and $\pi$. The score of a pair $(\pi, \sigma)$ on $m$ is then the score of the AUTUMN program it produces on completion.

$$\text{score}_m(\sigma, \pi) = \text{score}_m(\sigma(T, \phi)) \tag{3}$$

**Implementation** Computing marginal probabilities in AUTUMN programs is in general intractable, making computation of score functions a challenge. We approximate these scores using importance sampling.

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
