# OpenReview forum: "Causal Inductive Synthesis Corpus"
_NeurIPS.cc/2020/Workshop/CAP — NeurIPS 2020 CAP Workshop_

### Official Review · AnonReviewer1 · 2020-10-19
**Interesting dataset that has potential for interesting solutions**

**Rating:** 7
**Confidence:** 3

**Review:**

### Summary ###
The paper introduces CISC - a dataset of problems in the AUTUMN language (which is also introduced in the paper).
The dataset introduces an interesting set of RL-like problems that can be solved by correctly synthesizing an AUTUMN program.

This is an interesting dataset that serves as a good first step toward interesting solutions, and I thus vote for acceptance.

### Questions and Notes to Authors ###
L87: "Let L denote the set of all models" - what exactly is a "model"?

I miss some concrete examples for models and examples. I miss a connection between the image of the grid-world, the program, and the set of models. In Figure 1 I couldn't perfectly understand the code of the program and what are its possible trajectories.
I understand the page limitation, but I think that concrete examples will be more helpful for the reader, and maybe Section 2 could be moved to an appendix?

Figure 3 is great in providing more examples (although, note that the mouse cursor was captured in the images).

---

### Decision · Program_Chairs · 2020-11-02

**Decision:**

Accept

**Comment:**

Reviews were positive, so I recommend acceptance.